# Photobiomodulation as Medicine: Low-Level Laser Therapy (LLLT) for Acute Tissue Injury or Sport Performance Recovery

**DOI:** 10.3390/jfmk9040181

**Published:** 2024-09-27

**Authors:** Julia Lawrence, Karin Sorra

**Affiliations:** 1Arroscience Inc., Toronto, ON M2J 4R3, Canada; jlawrence@arroscience.com; 2Rotman School of Management, University of Toronto, Toronto, ON M5S 3E6, Canada

**Keywords:** low-level laser therapy, wound healing, sport performance, acute tissue injury, photobiomodulation

## Abstract

**Background/Objectives**: Low-level laser therapy (LLLT) has gained traction in sports and exercise medicine as a non-invasive therapeutic for preconditioning the body, exertion recovery, repair and injury rehabilitation. LLLT is hypothesized to modulate cellular metabolism, tissue microenvironment(s) and to decrease inflammation while posing few adverse risks. This review critically examines the evidence-base for LLLT effectiveness focusing on immediate care settings and acute/subacute applications (<6 months post-injury). **Methods**: A comprehensive literature search was conducted, prioritizing systematic reviews, meta-analyses and their primary research papers. **Results**: Findings are relevant to trainers and athletes as they manage a wide range of issues from superficial abrasions to deeper tissue concerns. LLLT parameters in the research literature include wide ranges. For body surface structures, studies show that LLLT holds promise in accelerating wound healing. In sport performance studies, LLLT is typically delivered pre-exercise and reveals beneficial effects on exertion recovery, improvements in muscle strength, endurance and reduced fatigue. Evidence is less convincing for acute, deep tissue injury models, where most studies do not report significant benefits for functional outcomes over conventional therapeutic modalities. **Conclusions**: Variability in LLLT delivery parameters and findings across studies underscores a need for clear treatment guidelines for the profession. Technical properties of laser light delivery to the body also differ materially from LED devices. Sport physiotherapists, team physicians, trainers and athletes should understand limitations in the current evidence-base informing photobiomodulation use in high-performance sport settings and weigh potential benefits versus shortcomings of LLLT use in the mentioned therapeutic contexts.

## 1. Introduction

Low level laser therapy (LLLT) is a clinical application of laser light to the body with aims to modulate tissue recovery and repair, decrease inflammation and/or reduce pain. LLLT stimulates cellular repair mechanisms and is calibrated such that intended therapeutic effects are in response to wavelengths of light and not due to heat. Introduced as a Medical Subject Heading (MeSH) term in 2002 [1], LLLT is used interchangeably with other terms, including photobiomodulation (PBM), laser biostimulation, cold laser and laser phototherapy [2]. LLLT’s therapeutic applications are wide-ranging and include managing musculoskeletal conditions, sports injuries, dental surgery, dermatologic procedures, general surgery, oncology and veterinary medicine [3]. Given such sweeping use of LLLT as a non-invasive and medication-free therapeutic, it is clinically relevant to ask probing questions of LLLT’s supportive research and experimental data. For instance, is LLLT use in immediate care settings, such as a “recovery window” therapeutic in sport performance or as a treatment for acute musculoskeletal injury, anecdotal, evidence informed, or evidence-based? In this paper, findings from the LLLT clinical literature are assessed to address this question. Is laser therapy’s widespread therapeutic use informed by high quality clinical evidence in acute settings?

Endre Mester is widely acknowledged as the “godfather” of LLLT by researchers and clinicians working in this space. This is in recognition of his unexpected finding in 1967 that low-power laser application to tissues could lead to hair growth and improved wound healing in mouse models [4]. Mester and colleagues went on to evaluate 15 biological systems, including human wound models with difficult to treat ulcers [5], and hypothesized that LLLT may have broad application and utility in human health and healing.

The FDA categorizes four classes of lasers (I to IV), plus three subclasses (IIa, IIIa, IIIb) [6], according to a laser’s power and risks with use. Class IIIb and IV are used in clinical and therapeutic settings. There are a growing number of FDA-, EU-, and UK-approved photobiomodulation devices, although not all devices sold on the market are necessarily approved for therapeutic use with human subjects. Under FDA regulations, a 510(k) application is required for new LLLT device clearance [7]. Market authorization by regulators includes biocompatibility data versus an established device, as well as clinical performance testing, software performance, and electrical and thermal safety testing. Not all marketed devices adhere to such standards.

Photobiomodulation or PBM is used as an umbrella term for laser light clinical treatments, as well as LED-based applications. There is an ongoing debate in both clinical (among practitioners) and commercial settings (manufacturers and distributors) about the bioequivalence or non-equivalence of functional outcomes with laser or LED devices [8]. The FDA classifies LEDs differently from laser diodes and therefore LED devices are not subject to Federal laser product performance standards [7].

In clinical practice settings, LLLT dosing is a key variable. The World Association for Photobiomodulation Therapy (WALT) is a leading organization supporting research, education, clinical application and treatment standards [9]. WALT publishes LLLT dose and administration recommendations for Class IIIb laser use in chronic arthritic and inflammatory conditions (780–860 nm or 904 nm GaAs nanosecond pulse lasers) [9].

The aim of this review is to assess the available data on LLLT effectiveness and safety, in immediate or acute settings of tissue recovery and repair. Findings are evaluated for laser use in settings relevant to athletes and trainers ranging from wound healing, athletic performance/recovery and acute musculoskeletal injury rehabilitation.

## 2. Literature Search Strategy

Therapeutic topics of interest and in-scope literature ranged from management of: (1) surface conditions such as wound healing (pre-, post-, and non-surgical), (2) sport performance/recovery models, to more complex sequelae including (3) acute/subacute soft tissue injuries (<6 months onset). The aims were to identify clear LLLT parameters and outcome measures to inform nonpharmacologic approaches that may support musculoskeletal health, functional recovery and to identify areas for future research. This review considers the breadth of scientific evidence from the existing published literature rather than depth with aims to identify gaps in the overall state of the research. By its nature, a review paper addresses broad questions and synthesizes concepts.

Out-of-scope topics included LLLT management of neurological conditions, dermatologic procedures, ophthalmology interventions, chronic pain, behavioural or mental health conditions, research with animal models and chronic injuries (>6 months onset). Other exclusion criteria for LLLT research included no mention of injury timelines, LED-only device research, non-musculoskeletal injuries and non-English language publications (Table 1). Retrieved papers were assessed to document alignment with the literature search inclusion and exclusion criteria.

Literature search criteria included meta-analyses, systematic and scoping reviews and primary studies informing on therapeutic laser use, specifically LLLT, in performance/recovery and acute tissue repair. Aims of the literature synthesis included informing therapeutic choice and shared decision-making for practitioners and their clients. Literature databases included PubMed, Google Scholar and Mendeley, with no timeline restriction up to 2023, although prioritizing publications within the previous 5 years. Research focused on laser device interventions or combined laser treatment clustered with LED technology. Studies using LED therapeutic protocols as the sole light intervention were excluded due to potential heterogeneity introduced across research studies with lower power devices and uncertain effects of LED on cellular constituents or deeper tissue structures. The topic of LED therapeutic use is evaluated elsewhere [10,11].

The quality of evidence described in the literature was categorized as follows: evidence was moderate-to-high quality if there were clear, prospective analysis plans with adequate sample size, treatment controls, statistical power, clear LLLT dosing and treatment parameters, and with clear data reporting. Evidence was deemed low quality when studies were constrained by unclear research methods, vague study population(s), small sample sizes and low statistical power.

## 3. Findings from the Clinical Literature

LLLT uses beams of light at specific wavelengths, typically red and near-infrared wavelengths, to stimulate cellular metabolism and modulate tissue microenvironments [12]. Class IV lasers are among the most powerful used in light therapy, while LED devices are considered to have shallower penetration depth compared to LLLT (Figure 1).

### 3.1. Wound Healing

Wound healing processes are relevant to trainers and athletes as they manage a wide range of issues from superficial abrasions to deeper tissue concerns that impact performance. Wound healing requires coordination of cellular and molecular events including cell migration, proliferation and extracellular matrix deposition (e.g., collagen). Scar formation is a natural stage of the healing process; however, excessive scarring may also result in functional impairments and impede tissue range of motion. Already 50 years ago, Mester et al. [5] proposed that laser light stimulates cells at the lips of a wound and in the wound bed accelerating healing. Clinically, this remains an unmet need today. Novel and adjunctive treatments are needed to reduce scarring, accelerate wound closure, improve wound esthetics, and improve outcomes. Does LLLT as a nonpharmacologic approach facilitate wound healing?

#### 3.1.1. Post Surgery

LLLT has been studied and is used clinically to accelerate healing of postsurgical (“fresh”) wounds and to minimize functional or traumatic sequelae. For the purposes of this paper, the focus is on clinical use of non-ablative lasers. Artzi et al. (2020) [13] published a systematic review on the utility of laser treatment on wound healing and scar mitigation in post-operative settings. Fourteen clinical studies met inclusion criteria, and included scar treatment following plastic surgery, breast, thyroid, or hernia surgeries. Pre- and post-operative laser treatment scores based on validated scales included the (1) Patient and Observer Scar Assessment Scale, (2) Vancouver Scar Scale and (3) Global Assessment Scale. Pulsed dye laser (PDL), carbon dioxide and diode lasers were the most frequently used devices in the studies followed by potassium titanyl phosphate (KTP) and erbium glass (Er-Glass) lasers. Various treatment protocols were used: most studies utilized three to four laser treatments at intervals of 2–4 weeks with variable wavelengths applied from 585–596 nm or 810–830 nm with select studies utilizing much longer wavelength light. In these post-operative settings, LLLT facilitated healing and improved scar appearance versus no laser controls. Artzi (2020) [13] recommend early (upon suture removal), post-op use of laser to support the wound healing process. Diode lasers, PDL and carbon dioxide lasers were reported to have the best effects relative to controls. During LLLT, wounds are monitored for side effects including redness, discoloration, pain and infection. Seago et al. (2019) [14] in an International Consensus Guideline on postsurgical and traumatic scarring, state laser treatment improves wound appearance, pliability and tissue range of motion, and that laser therapy should be considered a primary therapeutic option for anticipatory scar mitigation, as well as scar related pain and pruritus. A recently published systematic review and meta-analysis of 12 randomized, controlled trials further informs this topic [15]. Future controlled studies should continue to guide athletes and sports medicine specialists and inform standardized treatment protocols in these settings.

#### 3.1.2. Pre-Conditioning of Wounds

In non-sport settings, pre-conditioning physical therapy is performed before orthopedic surgeries to strengthen muscles and improve range of motion around an affected joint and post-operatively to help reduce pain, swelling, and accelerate overall functional improvements and mobility. Similarly, LLLT pre-treatment of a surgical site may be beneficial to patients’ healing. Of note, Friedman 2020 [16] studied laser pre-treatment of surgical incision sites for individuals undergoing planned lipoma removal. Subjects served as their own controls. Treatment included one pass with an Er-Glass laser (50% overlap between adjacent laser grids) with energy level 40 mJ, 15 ms pulse duration at wavelength 1540 nm, 24 h before surgery. This “conditioning” approach of the skin resulted in significant improvement of final scar appearance 12 months post-procedure. Further, high quality research is needed to generate guidelines for practice.

### 3.2. Athletic Training and Performance

Innovation in sport technologies aim to optimize performance and recovery for amateur and professional athletes. LLLT is used in select athletic settings across various treatment parameters and timelines. Studies with healthy volunteers (uninjured ≥ 3 months) were chosen to limit interference/confounding factors of other training regimens. Twelve systematic reviews and meta-analyses published 2019–2023 were retrieved in the literature search (Table 2) [17,18,19,20,21,22,23,24,25,26,27,28]. Across the 12 papers, 108 studies were assessed with 64 studies of interest on recovery and performance measures in athletic settings. Fifty-two of the 64 studies concluded benefits in performance with LLLT regimens when accompanying training for younger, healthy participants/athletes. Studies were interpreted with positive outcomes based on conclusions by each research group as published via peer review process. The most often used LLLT parameters included 808–850, or 905 nm wavelength of light, less than 30 s exposure per point, and <10 J of energy over 17 or fewer exposure points. Beyond the most used ranges, LLLT parameters spanned 632.8–980 nm, up to 10 min exposure per point, and <60 J over at most 85 exposure points.

Outside a sports context and not listed in Table 2, four studies were assessed that asked questions on LLLT use and aging and included older participants (women 60–70 years) [29,30,31], or people with heart failure (35–65 years of age) [32]. Across the four studies, improvements in functional capacities were reported with LLLT (e.g., increased repetitions of exercises such as leg flexion/extensions) [29,30,31], and delayed fatigue onset [30,32].

González-Muñoz et al., (2023) [17] evaluated 15 RCTs published 2017–2022, assessing LLLT in sports performance (Table 2). From the 15 studies, 9 included LLLT [33,34,35,36,37,38,39,40,41], 8 reported positive functional outcomes such as increased time on the soccer pitch, longer time to reach exhaustion, and accelerated muscle recovery [33,34,35,36,37,38,39,40]. Variable laser parameters were used, with energy density ranging from 0.285–30 J per treated point (2–17 or 85 exposure points), with treatment immediately to 40 min before [35,36,37,38,40], after [33] or a combination of before and after [34,39] exertion. There is one exception of immediate treatment with performing LLLT at 3-, 6- and 24-h post exertion timepoints [38]. González-Muñoz et al. [17] suggest future research is needed to specify precise LLLT parameters, as well as conducting RCTs that evaluate professional athletes and untrained healthy individuals to further understand the utility of LLLT for both high-performance sport versus “weekend warriors”.

Oliveira et al. (2023) [18] assessed 15 RCTs published 2012–2018 on LLLT’s impact on skeletal muscle performance and recovery with all 13 studies [34,36,42,43,44,45,46,47,48,49,50,51,52] using LLLT reporting favorable results for athletic performance, and inflammatory marker expression (Table 2). LLLT promoted maximum voluntary contraction, peak strength, isometric capacity and longer time to achieve fatigue/exhaustion. Nine studies utilized LLLT before exercise [36,43,44,45,46,48,49,51,52], two after exercise [34,42], and two using a combination of before and after exercise [47,50] with 640–670 nm, 800–880 nm or 905–980 nm, 0.095–50 J per point (3–17 or 42 exposure points) for a total of 10–480 J total energy per limb, and application times ranging < 30 s up to 6 min. Oliveira et al. [18] concluded that LLLT had favorable results in muscle regardless of treatment timeline, although better results were reported when applying LLLT prior to exercise.

Dutra et al. (2022) [28] performed a systematic review and meta-analysis on the ergogenic effects of LLLT in sport specific exercises. From 37 included studies, 24 involved laser therapy [33,42,43,44,46,49,52,53,54,55,56,57,58,59,60,61,62,63,64,65,66,67,68,69], 20 with positive results [33,42,43,44,46,49,52,53,54,55,56,59,60,61,62,63,64,67,68,69] (Table 2). Dutra et al. (2022) were specifically interested in preconditioning LLLT in sport. The LLLT parameters across the assessed studies ranged from 600, 780–905 nm, 0.81–60 J per point across 2–29 points of exposure for a total of 12–540 J over 13–300 s of exposure. The findings suggest LLLT increases muscle endurance in single-joint exercises, and time to exhaustion in cycling. However, no significant effects in muscle strength in single-joint exercises, running or swimming performance scores were reported.

Luo et al. (2022) [19] conducted a meta-analysis on performance and soreness recovery for athletes. From 24 included studies, 16 involved laser therapy [33,36,42,43,45,50,58,59,60,61,70,71,72,73,74,75], 14 with positive results [33,36,42,43,45,50,59,60,61,70,71,72,73,74] (Table 2). All but three used pre-exercise LLLT, one applied LLLT after exercise [50], and one used pre- and post-exercise LLLT [71]. Parameters were wide ranging, with wavelengths of 633, 655, 810–905 nm, 0.285–45 J energy per treated point (2–9, 17, 42, 85 points) and 15–600 s exposure. Overall, LLLT was found to improve muscular performance and enhance soreness recovery. Luo et al. [19] recommend LLLT use before or after competition to regain muscle capacity between strenuous work.

De Marchi et al. (2022) [20] evaluated eight RCTs published 2011–2022 to assess whether LLLT impacts exercise-induced oxidative stress among healthy individuals. Five in-scope studies [33,36,42,49,76] evaluated lower body LLLT application, three treating pre-exercise [36,42,49], one post-exercise [33], and one using a combination [76]. The parameter ranges were 810, 830 or 905 nm, 0.285–30 J per point (6–17 or 85 points), and 25–228 s exposure. Outcomes of 5 studies [33,36,42,49,76] showed LLLT was beneficial in both performance and recovery (Table 2).

In a 2021 narrative review by Ailioaie et al. [21], 39 studies evaluated LLLT’s impact on sports performance; 23 included LLLT, and the remaining 16 were LED-only studies. Eighteen of the 23 LLLT studies reported “valuable protective and ergogenic effects” of laser treatment for clinically healthy and physically active individuals [33,36,37,38,39,42,43,45,48,50,62,63,64,70,77,78,79,80] (Table 2). Treatment timelines varied across LLLT with 12 studies investigating LLLT use before exertion [33,36,38,42,43,45,48,62,63,64,70,80], 3 after [37,50,77], and 3 used a combination of before and after exertion [39,78,79]. LLLT parameter ranges were 640 nm, 810–905 nm, < 60 s or 180–381 s, and 0.095–50 J per treated point (3–9, 17, 85 points). Taken together, LLLT was reported to be a key companion therapeuetic for high-performance sport and recovery. Ailioaie et al. [21], also commented on studies that were inconclusive demonstrating no effect of LLLT. Technological limitations (devices, techniques and parameters used), heterogeneous study participants, or unclear protocols for physical activity were cited as some of the challenges [21].

In a scoping review, Alves et al. 2019 [22] evaluated the “immediate” effects (within 40 min of exertion) of LLLT on muscle performance across 27 studies with healthy individuals. A total of 25 studies used laser [29,30,31,32,36,45,47,49,50,52,58,59,60,61,67,68,69,73,79,81,82,83,84,85,86], of which 21 showed improved markers of muscle performance and endurance (assessments: best task performance, cardiorespiratory assessment, increased exercise load or number of repetitions studies) [29,30,31,36,45,47,49,50,52,59,60,61,67,68,69,73,79,81,82,83,85] (Table 2). From these studies, 16 studies treated subjects prior to exertion (immediately to 10 min prior) [30,31,45,49,52,59,60,61,67,68,69,73,78,81,83,85], 4 studies treated post-exercise (immediately to 40 min after) [29,47,50,82], and 1 treated individuals between exercise sets [79]. Parameter ranges were 655, 660, 808–950 nm, 8–300 s exposure, and 0.6–50 J per treated point (2–12, 29, 30, 42 points). The doses per point were 7 J and 30 J. Alves et al. [22] concluded that LLLT optimizes muscle performance and reduces fatigue, especially when administered pre-exertion.

Vanin et al. (2018) [23] retrieved 39 RCTs published prior to March 2017 with the aim of determining optimal LLLT dosages impacting athletic performance and muscular fatigue. A total of 27 included laser [43,45,47,48,49,50,52,58,59,60,61,64,67,68,71,73,77,78,79,81,82,83,85,86,87,88,89], and 24 reported positive results [43,45,47,48,49,50,52,59,60,61,64,67,68,71,73,77,78,79,81,82,83,85,87,89]. Vanin et al. [23] revealed positive results for small muscle groups within the LLLT energy dose range of 20–60 J total, and 60–300 J total for larger muscles. From the studies with favorable outcome, 16 studies treated participants pre-exertion [43,45,48,49,52,59,60,61,64,67,68,73,81,83,85,89], 4 treating post-exertion [47,50,77,82], and 4 using a combination of pre- or post-, or in-between exercise sets [71,78,79,87] (Table 2). The results suggest the use of LLLT may improve athletic performance and reduce muscle fatigue, with best effects occurring when used before exertion.

Ferraresi et al. (2016) [24] retrieved 46 RCTs published prior to 2016. A total of 34 included LLLT intervention [29,30,31,43,45,47,48,49,50,51,52,58,59,60,61,64,67,68,69,71,73,77,78,79,81,82,83,84,86,87,88,89,90,91] and 28 reported positive results [29,30,31,43,45,47,48,49,50,51,52,59,60,61,64,67,68,69,71,73,77,78,79,81,82,83,87,89] (Table 2). Across healthy, athletically trained or untrained individuals and elite athletes, the results showed LLLT favorably affected muscle performance and recovery. Dose recommendations were categorized by activity, and treatment timeline. A therapeutic window for smaller muscle groups (e.g., biceps brachii) was identified as 20–80 J energy dose. Larger muscles (e.g., quadriceps femoris) responded favorably when treated with total energy dose range of 56–315 J. From these results, there appears to be favorability of using LLLT pre-exertion with 19 included studies pre-exertion [30,31,43,48,49,51,52,59,60,61,64,67,68,69,73,78,81,83,89], 5 post-exertion [29,47,50,77,82], and 4 either incorporating both pre- and post-exertion or in between sets [71,78,79,87]. Further research on how different muscle types, groups and sizes respond to LLLT will provide clinicians with data on how to customize treatments.

Three additional systematic reviews or meta-analyses published earlier than those discussed above [25,26,27] had similar research questions and with overlapping coverage of the primary research so are not re-reviewed here.

Across all 12 systematic reviews, LLLT results in beneficial outcomes for exertion and recovery although the quality of evidence of the primary papers is considered low (Table 2; Figure 2). LLLT administration in sport performance settings requires additional high-quality evidence. Reproducible results are required to gain better insight on how to specify treatments for optimal muscle endurance and performance. LLLT shows promise to impact sport performance and improve recovery. LLLT as an adjunctive therapeutic is gaining attention with athletes, sport franchises and federations, in particular on the use of LLLT for pain management in sport by the International Olympic Committee [92]. As of the date of this literature synthesis, no peer-reviewed, evidence-based guidelines of laser use in athletic performance, support or recovery were found.

### 3.3. Management of Acute Musculoskeletal Injury

Here, the focus is on acute/subacute (<6 month) injuries, where the literature included lateral epicondylitis (LE), rotator cuff/bicep tendinitis or subacromial impingement syndrome (SIS) (Table 3). Functional outcomes across studies included: Disabilities of the Arm, Shoulder, and Hand (DASH), Quick DASH, the Shoulder Pain and Disability Index (SPADI), and the Shoulder Disability Questionnaire (SDQ), as well as pain measurement scales (Visual Analog Scale, VAS). From the analysis of the literature, there are many fewer publications focusing on LLLT utility in acute/subacute injuries (Table 3) versus those mentioned above in sport performance models (Table 2).

Tripodi et al.’s., (2021) systematic review [93] evaluated 17 injury studies, with 5 acute/subacute studies for individuals with LE [98,99,100], tendinitis [101], or SIS [102] (Table 3). Emanet et al. (2010) [98] used WALT guidelines [9] to inform their treatment protocols for LE (1 J/cm^2^ for 2 min at 100 W peak power). Significant improvements in DASH and patient-reported questionnaires were reported at 3- and 12-weeks. Kaydok et al. (2020) [99] used LLLT (904 nm, 2.4 J/cm^2^ for 30 s) as an active control versus high level laser therapy (protocol 1: 1064 nm, 6 J/cm^2^, 8 W for 75 s; protocol 2: 1064 nm, 120–150 J/cm^2^, 6 W for 30 s). Both intensities significantly improved LE functional assessments (Q-DASH and hand grip strength), as well as VAS after 3 weeks of treatment. Lam et al. [100] randomly assigned 39 LE patients to receive either LLLT (904 nm, 2.4 J/cm^2^, for 11 s) or a sham laser. Their results found LLLT effective in improving functional assessments (DASH and maximum grip strength). For rotator cuff tendinitis, Eslamian [101] found significant improvements in function (based on SDQ) at a 3-week assessment, with no noted advantages for range of motion. Bal et al.,’s (2009) [102] SIS study reported no benefit of LLLT plus a home exercise program versus a home exercise program alone. They used an energy dose 20% below WALT recommendations (1.6 J/cm^2^ versus minimum 2–3 J/cm^2^), which may explain the study reporting no benefit of LLLT versus exercise [102]. Taken together, the five studies are largely using subjective test measurements (e.g., DASH, Q-DASH, SDQ). Objective measurements (e.g., hand grip strength, and range of motion) showed statistically significant improvements in two of the five studies [99,100].

Awotidebe et al. (2019) [95] evaluated 11 RCTs with clinical or radiological diagnoses of various shoulder disorders (SIS and rotator cuff tendinitis RCTs within the acute/subacute injury timeline). The study selection criteria required the intervention group to receive exercise and either class IIIb (780–806 nm with 5–500 mW) or class IIIb (904 nm with >5 mW) with the 2010 WALT [103] recommendations. Four studies [101,102,104,105] recruited subjects with acute/subacute injuries (<6 months), three of those studies [102,104,105] found no advantage to incorporating LLLT while one [101] reported LLLT superior to routine physiotherapy (solely based on subjective measures). Awotidebe et al. [95] concluded that LLLT combined with an exercise program was no more effective than exercise alone.

Taylor et al. (2020) [94] set out to identify LLLT dosing variables for neuromusculoskeletal conditions. From 86 included studies, 4 [100,101,102,105] covered acute/subacute injuries, with 3 [100,101,102] previously referenced by Tripodi [93], and 1 [105] by Awotidebe [95]. Taylor’s scoping review synthesized findings from individual trials across various performance measures, acute and chronic injuries, neuromuscular and neurological conditions. While general treatment approaches are summarized in their review with LLLT’s clinical value being noted, the heterogeneity of study participants and treatment modalities make it challenging to draw firm conclusions on recommendations for acute injury settings.

Two other review articles assessed acute injury models [96,97]. Haslerud [97] focused on shoulder injuries, and from four in-scope studies [101,102,104,105], it was reported that LLLT does not contribute meaningful benefits as an adjunct to conventional therapies. For Clijsen [96], while predominantly a review on LLLT’s impact on pain, they did note two studies using functional measures, one study in LE [98] and another in carpal tunnel syndrome [106]. However, neither found benefit in implementing LLLT with functional assessments.

Overall, there are few published works focusing on LLLT’s clinical impact on acute/subacute injuries (Table 3, Figure 3). Across the five review articles [93,94,95,96,97] published 2015–2021, there was minimal to no additional benefit when incorporating LLLT with conventional treatment plans (e.g., heat therapy or physiotherapy; Figure 3). Across the 7 primary studies [98,99,100,101,102,105,106], all use Class IIIb lasers as opposed to Class IV. An argument could be made that the Class IIIb laser might deliver dosages below what is required for therapeutic impact on acute injuries. Eslamian study [101], was included in four of the review papers [93,94,95,97], and while concluding LLLT had beneficial results, it is important to note the numerous interventions used in combination with laser including: heat, ultrasound, electrotherapy, or trans-cutaneous electrical nerve stimulation (TENS). These multimodal therapeutics likely recruit healing pathways, obscuring potential treatment impact of LLLT alone. Statistically significant (*p* < 0.05) beneficial results were noted solely in subjective measures. As the literature stands, LLLT does not appear to offer meaningful therapeutic benefit(s) in acute injury tissue rehabilitation, other than perhaps recruiting analgesic pathways.

### 3.4. LLLT at a Cellular Level

The light-sensitive photoreceptor or “chromophore” for LLLT (activated by red to near-infrared light; Figure 1) is proposed to be mitochondrial cytochrome C or, alternatively, mitochondrial bound water [107,108]. When tissue is damaged, overexerted or otherwise dysfunctional, ATP production decreases. In the presence of red or infrared light, mitochondrial function increases, ATP production increases, and the release of reactive oxygen species goes up which may act in molecular signaling, as well as nitric oxide to tip cellular signaling pathways in favor of anti-inflammatory mediators [12,109] (see Figure 4). The activation of transcription factors, such as NFkB, regulate cell migration, proliferation, inflammatory and stress-induced responses, and hypoxia-induced factor (HIF) activation upregulates several genes and glycolysis enzymes, which allows ATP synthesis in hypoxic conditions in an oxygen-independent manner.

Collagen is the major protein in the extracellular matrix that constitutes most of the connective tissue during wound healing. In vitro studies show that LLLT increases cell viability, cell migration, proliferation and collagen synthesis [12,109]. LLLT induces macrophages to release factors that stimulate fibroblast proliferation. In addition, LLLT promotes the production of interleukin-1 alpha (IL-1α) and interleukin-8 (IL-8), which stimulate the migration of keratinocytes. LLLT also increases the expression of platelet-derived growth factor (PDGF) and transforming growth factor-β (TGF-β). PDGF stimulates mitogenicity and chemotaxis of fibroblasts and smooth muscle cells and chemotaxis of neutrophils and macrophages.

LLLT thereby recruits tissue recovery, healing and/or repair mechanisms [12] as ATP levels and related metabolic processes are activated. The cellular microenvironment tips in favor for recovery and repair.

## 4. Discussion

The objective of this review was to critically assess the LLLT literature in immediate care settings, for sport performance exertion or acute injury rehabilitation. Across these distinct therapeutic settings, each have their own complexities associated with the recovery process, spanning from superficial trauma to recovery of deeper tissue targets. Differences exist in technical specifications between Class IIIb or Class IV laser LLLT and LED devices (Table 4). The therapeutic choice between laser or LED use will depend on specific therapeutic goals, target tissue depth and desired treatment outcomes for the light therapy.

PBM parameters reported in the literature show wide ranges as summarized in Table 4. For sport training contexts, there is no standardization of total light energy delivered to muscles, energy density (J/cm^2^), ideal power output (mW) of the laser device or time/duration of light exposure to target tissue(s) across research studies. One reason contributing to these inconsistencies is a wide range of PBM devices available on the market both regulated and unregulated. Laser devices, LED and mixed array devices are available to practitioners and researchers all contributing variation to the understanding of efficacious treatment protocols.

Wound healing, however, is an injury model that avails itself to assess healing time course with LLLT using objective outcomes such as visual tracking and direct measures of healing. In research studies, wound healing has the unique benefit of being able to use the same incision or ulcer as a control and treatment area [13]. This eliminates confounding factors that arise when attempting to match control subjects and experimental subject conditions (e.g., wound depth, body composition, skin color, etc.). Advanced therapeutic options for large or complex wound closures remain limited resulting in prolonged healing times and increased risk of infection. However, the available data suggest that laser treatment improves wound appearance and tissue integrity, and that laser therapy should be considered a primary therapeutic option. The most recent International Consensus (2020) deems first-line laser treatment for traumatic scars and contractures as the best available care [14]. Skin color and body composition affects the properties by which light interacts with tissue [112]. Thus, additional consideration is required for treatment protocols to calibrate LLLT approaches according to skin color and/or adiposity. In the future, laser therapy could be tested as an adjunctive treatment to facilitate engraftment of bioengineered skin substitutes with advanced dressings, or stem cells to accelerate skin and tissue repair.

In sports performance and recovery models, LLLT demonstrated predominantly positive outcomes across individual research studies (see Figure 2). For studies that were inconclusive, short treatment cycle or significantly lower energy administered may have impacted the outcomes. From the included reviews [17,18,19,20,21,22,23,24,25,26,27,28], there is a wide range of LLLT parameters applied across various therapeutic settings (Table 2). One of the inconclusive studies [88] referenced WALT in their choice of LLLT dose, delivering 2.5–3.5 J/cm^2^ at 5 and 10 min, respectively. While WALT is a consensus guideline on Class IIIb laser use for management of chronic conditions, the guidelines do not provide evidence-based guidance on LLLT use to manage exertion or acute muscular strain. Interestingly, a 2008 pilot study [88] of muscular fatigue in healthy male volunteers applied WALT’s recommendations as treatment protocols. Many studies [29,30,32,33,34,36,42,43,45,48,49,50,58,59,60,61,67,69,71,73,74,76,77,78,81,86,113], both favorable and inconclusive, explored LLLT’s impact on muscle damaging mediators such as creatine kinase and blood lactate. Such measurements allow for more objective, biomarker-based testing, and the findings suggest possible benefits of LLLT use for performance athlete recovery. Bettleyon and Kaminski [114] note micro-CK level elevations hinder the recovery process via cellular membrane damage, localized hypoxia, and electrolyte imbalances. One hypothesis is LLLT may reduce these mediators. Clearly, more translational research is needed on correlatives between muscle biomarkers and performance measures in the context of LLLT.

For other tissue types, few human studies are published on bone repair and LLLT to inform the athletic setting. At present, the bone literature is seemingly limited to mechanistic and animal models. A 2023 systematic review [115] was published evaluating human studies and the effect of LLLT on bone repair by assessing in vitro targets. To date, the underlying mechanisms by which LLLT might modulate bone formation remain unknown, though would serve to inform orthopedics in sports in the instance of fractures. Factors such as wavelength, energy density, exposure and frequency of LLLT might influence calcium signaling and the cellular mechanisms of bone repair and requires further research.

When comparing studies of LLLT in sport performance models (Figure 2) with those in acute injury settings (Figure 3), the number of LLLT research papers was far fewer in the setting of acute injuries. This could be due to challenges with funding and conducting studies in acute injury settings (e.g., subject recruitment) or may also reflect publication bias in which studies of LLLT in acute settings with inconclusive or negative results for LLLT intervention are not submitted or accepted for publication.

The effectiveness of introducing LLLT across all aspects of an athlete’s training and performance routine remains an open question. There are still limitations in the current evidence-base informing LLLT in sports settings. Additional variables such as athlete body composition, skin color, age and gender may all influence the therapeutic index of LLLT and desired outcomes. With laser therapy continuing to make headway in the sporting world, those planning to integrate this modality into their current routine should investigate how current limitations balance against potential benefits. Continued interest and research in this non-invasive therapeutic will continue to grow the database of knowledge and fine tune parameters for optimal results.

## 5. Conclusions

LLLT has emerged as a promising technology across the various aspects of an athlete’s training, performance, and recovery. From our investigation, LLLT interventions, both alone and as adjunctive therapy, yield variable outcomes in immediate care settings with beneficial effects for surface lesions/wound healing and athletic exertion/recovery. In contrast, the research suggests LLLT is of questionable utility for accelerating repair of deeper musculoskeletal structures in the setting of acute/sub-acute injury in the near-term post-injury period. LLLT’s effectiveness in supporting preconditioning, exertion and recovery is relevant for high performance sport This non-invasive modality may support quicker return-to-play times for athletes by mitigating fatigue and strain, allowing for more consistent training and performance. While subjective testing in the research studies is an important element of participants’ experiences, quantitative objective testing is needed to advance the field. Understanding how body composition, skin color, age and gender impact LLLT clinical outcomes is also needed. High-quality research must continue to inform the science and practice parameters of LLLT’s effects on functional outcomes. A call to action is the publication of guidelines for the profession to inform laser care alongside traditional therapeutic or conditioning protocols in sport and could be modeled after guidelines in other fields such as the International Consensus Guideline on Postsurgical and Traumatic Scarring [14] or WALT [9]. Such steps will serve to advance effective, safe and ethical use of PBM devices and advocate for improved regulations and standardization in this field for sport physiotherapists, team physicians, trainers and athletes.

## Figures and Tables

**Figure 1 jfmk-09-00181-f001:**
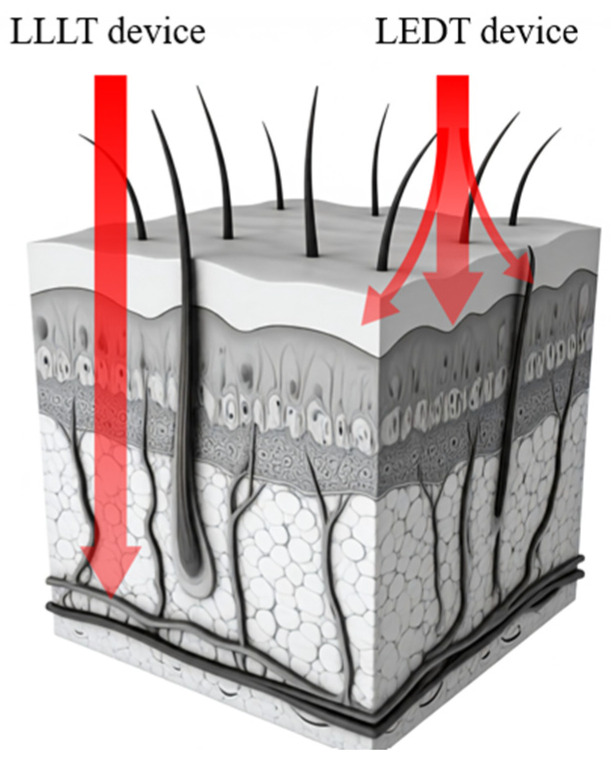
Low-level laser therapy (LLLT) and light-emitting diode therapy (LEDT) are both forms of red-light therapy used for various medical purposes, including the treatment of superficial wounds and sore or injured connective tissues. Not to scale; for illustrative purposes only.

**Figure 2 jfmk-09-00181-f002:**
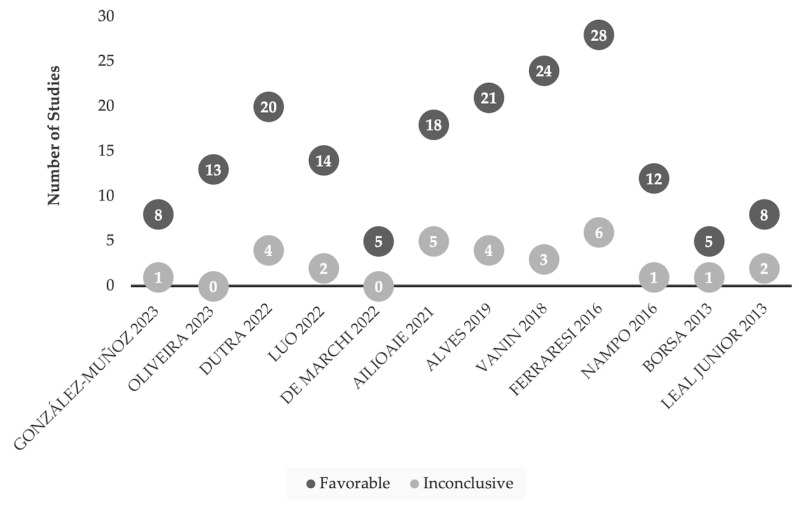
Comparison of favorable and inconclusive research studies of LLLT for sports-related models [17,18,19,20,21,22,23,24,25,26,27,28]. Research findings are considered favorable based on the conclusions drawn from the authors of LLLT use in sport performance settings (i.e., number of repetitions, time on field), or recovery (i.e., delayed onset of muscle soreness (DOMS)).

**Figure 3 jfmk-09-00181-f003:**
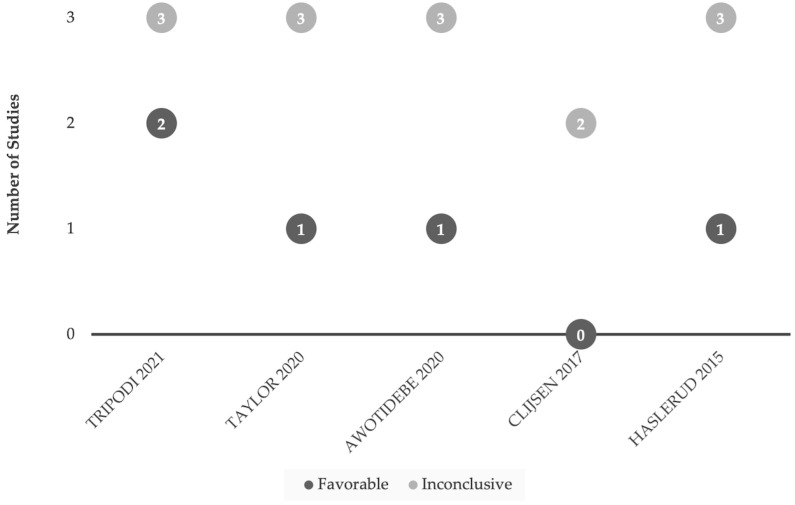
Comparison of favorable and inconclusive results when applying LLLT (Class IIIb) for acute or subacute (<6 month onset) injuries. Research findings are considered favorable based on the conclusions drawn from the authors [93,94,95,96,97]. Our interest was in studies emphasizing functional outcomes versus studies focusing primarily on subjective measures or pain management.

**Figure 4 jfmk-09-00181-f004:**
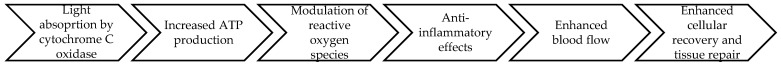
Delivery of red or infrared light to body structures may involve an interplay of biological processes that lead to decreased inflammation and accelerated recovery of strained or damaged tissues [12].

**Table 1 jfmk-09-00181-t001:** In scope and out-of-scope criteria for literature assessment.

Literature Search In-Scope Criteria	Literature Search Out-of-Scope Criteria
Systematic reviews and their primary papers Meta-analyses Scoping reviews Therapeutic laser use in acute/subacute settings or injuries (<6 months duration)	Sources not in the English language LED use only Animal models Chronic pain trials, neurology, dermatology, ophthalmology, behavioral, mental health, injuries beyond musculoskeletal (i.e., slipped disc) with neurological involvement, bone repair or veterinary medicine models No clear mention of injury timeline Chronic injuries (>6 months duration)

**Table 2 jfmk-09-00181-t002:** Summary of the technical features of LLLT in athletic performance-related research studies.

Review	RCTs Evaluated	Laser Parameter Ranges	Outcome	LLLT Pre-, Post-Exertion or Both	Quality of Primary Research Evidence
González-Muñoz 2023 [17]	15 total RCTs 9 laser/laser + LED 6 OOS	810, 850, 905 nm 0.285–30 J/point; 30–60, 510, 850 J total 32–228 s 2–17, 85 exposure points	8 of 9 favorable results	Pre-Exertion: 5 Post-Exertion: 1 Both: 2	Low
Oliveira 2023 [18]	15 total RCTs 13 laser/laser + LED	660, 800–980 nm 0.095–50 J/point; 10–60, 180–480 J total 16–30 s–<6.5 min 3–17, 42 exposure points	13 of 13 favorable results	Pre-Exertion: 9 Post-Exertion: 2 Both: 2	Low
Dutra 2022 [28]	37 total RCTs 24 laser/laser + LED 13 OOS	660, 780–905 nm 0.81–60 J/point; 12–84, 100–540 J total 13–300 s 2–29 exposure points	20 of 24 favorable results	Pre-Exertion: 20	Low
Luo 2022 [19]	24 total RCTs 16 laser/laser + LED 8 OOS	632.8, 655 810–905 nm 0.285–45 J/point; 12–60, 135–405 J total 15–600 s 2–9, 17, 42, 85 exposure points	14 of 16 favorable results	Pre-Exertion: 12 Post-Exertion: 1 Both: 1	Low
De Marchi 2022 [20]	8 RCTs total 5 laser/laser + LED 3 OOS	810–905 nm 0.285–30 J/point; 30–68, 180, 300, 360 J total 25–228 s 6–17, 85 exposure points	5 of 5 favorable results	Pre-Exertion: 3 Post-Exertion: 1 Both: 1	Low
Ailioaie 2021 [21]	39 RCTs (human); 22 animal studies 23 laser/LED 16 OOS	640, 810–905 nm 0.095–50 J/point; 10–60, 180–540 J total 16–381 s 3–9, 17, 85 exposure points	18 of 23 favorable results	Pre-Exertion: 12 Post-Exertion: 3 Both: 3	Low
Alves 2019 [22]	27 RCTs 25 laser/LED 2 OOS	655–660, 808–950 nm 0.6–50 J/point; 1.92–60, 180–300 J total 8–300 s 2–12, 29, 30, 42 exposure points	21 of 25 favorable results	Pre-Exertion: 16 Post-Exertion: 4 Both: 1	Low
Vanin 2018 [23]	39 RCTs 27 laser/LED 12 OOS	660, 810–905 nm 0.285–50 J/point; 4–60, 180–360 J total 10–300 s 2–17, 30, 42 exposure points	24 of 27 favorable results	Pre-Exertion: 16 Post-Exertion: 4 Both: 4	Low
Ferraresi 2016 [24]	46 RCTs 34 laser/laser + LED 12 OOS	660, 780, 808–950 nm 0.095–30 J/point; 4–60, 132–380 J total 10–720 s 1–7, 29, 42 exposure points	28 of 34 favorable results	Pre-Exertion: 19 Post-Exertion: 5 Both: 4	Low
Nampo 2016 [25]	16 RCTs 13 laser/laser + LED 3 OOS	655, 660, 808–970 nm 0.6–30 J/point, 12–60, 180, 360 J total 20–240 s 2–12, 42 exposure points	12 of 13 favorable results	Pre-Exertion: 11 Both: 1	Low
Borsa 2013 [26]	10 RCTs 6 laser/laser + LED 4 OOS	655, 810, 830 nm 3–40 J/point; 20–60 J total 30–100 s 2–5 exposure points	5 of 6 favorable results	Pre-Exertion: 5	Low
Leal Junior 2013 [27]	13 RCTs 10 laser/laser + LED 3 OOS	655/660, 808/810, 830 nm 0.6–30 J/point, 12–60, 360 J total 30–600 s 2–12, 42 exposure points	8 of 10 favorable results	Pre-Exertion: 7 Post-Exertion: 1	Low

J, joules; nm, nanometers; OOS, out of scope (LED-only study, exclusively pain model, non-English publication); RCT, randomized control trial. There is wide variability in energy values (J per point and overall total) based on how the primary research authors presented their data. Some reported the laser or LED diode as individual components, while others are combined. In evaluating the impact of lasers, where possible, the energy dosage was calculated based on solely the laser contribution. The total energy (J) listed, and number of exposure points are per limb (upper or lower).

**Table 3 jfmk-09-00181-t003:** Summary of the main features of LLLT and acute injury review papers.

Review	RCTs Evaluated	Participant Injury	Laser Parameters	Outcome	Quality of Primary Research Evidence
Tripodi 2021 [93]	17 RCTs 5 acute/subacute studies	Later Epicondylitis Rotator Cuff Tendinitis Subacromial Impingement Syndrome	830, 904/905 nm 1–4 J/cm^2^ 11–30 or 120 s	2 of 5 favorable results	Low
Taylor 2020 [94]	86 RCTs (1 SR, 1 MA) 4 acute/subacute studies	Lateral Epicondylitis Rotator Cuff Tendinitis Subacromial Impingement Syndrome	830, 904/905 nm 2–4 J/cm^2^ 11, 20, 90, 120 s	1 of 4 favorable results	Low
Awotidebe 2020 [95]	11 RCTs 4 acute/subacute studies	Rotator Cuff Tendinitis Subacromial Impingement Syndrome Unilateral Shoulder Pain	830, 904 nm 1.6–4 J/cm^2^ 20, 60, 90, 120 s	1 of 4 favorable results	Low
Clijsen 2017 [96]	18 RCTs 2 acute/subacute studies	Lateral Epicondylitis Carpal Tunnel Syndrome	830, 905 nm 1 J/cm^2^ 1 or 2 min	0 of 2 favorable results	Low
Haslerud 2015 [97]	17 RCTs 4 acute/subacute studies	Rotator Cuff Tendinitis Subacromial Impingement Syndrome Unilateral Shoulder Pain	830, 904 nm 2–4 J/cm^2^ 20, 60, 90, 120 s	1 of 4 favorable results	Low

MA, meta-analysis; nm, nanometers; RCT, randomized control trial; SR, systematic review.

**Table 4 jfmk-09-00181-t004:** Comparative parameters of low-level laser therapy (LLLT) and light emitting diode (LED) devices for musculoskeletal therapeutic use.

Parameter	Low-Level Laser Therapy (LLLT)	Light Emitting Diode (LED)
**Light Source**	Light Amplification by Stimulated Emission of Radiation (LASER). Uses a gas or crystal medium to produce light; electricity passes through the medium causing electrons to emit photons that are amplified Laser diode: LASER; Class of lasers that generate laser radiation through a semiconductor; stimulated emission	LED from spontaneous emission. Use semiconductor materials and much simpler p-n junctions than laser diodes, to produce non-coherent (less directional), broad spectrum light; the process is called electroluminescence
**Classification**	Class IIIb: treatment of musculoskeletal conditions (e.g., sport performance, acute and chronic inflammatory conditions), would healing Class IV: similar to Class IIIb above but also deep tissue repair and rehabilitation	N/A
**Emitted Light**	Monochromatic	Broad spectral width (~5% of central wavelength) [110]
**Coherency**	Coherent light	Non-coherent light
**Wavelength**	632.8–660 & 808–980 nm *^,†^	630–980 nm [24,26]
**Depth of Penetration**	Both superficial and deeper structures; suggested to penetrate ≤50 mm [111]	Superficial; 2–10 mm [11]
**Energy Density**	3–500 & 1071.43–1785 J/cm^2^ *^,†^	1.5–8 J/cm^2^ [24,26]
**Exposure Time**	8–720 s *^,†^	30–360 s [24,26]
**Power Output**	Class IIIb: 5–500 mW; Class IV: >500 mW	10–300 mW [24,26]
**Skin Application**	Direct contact or non-contact protocol for wounds	Direct contact or non-contact protocol for wounds
**Other features**	Wavelength specificity, in-clinic/professional treatments, targets treatment to damaged tissue	Lower cost device, possible at-home administration, may expose larger areas to treatment

* Based on findings from the sport performance literature. See Table 2. ^†^ Based on findings from the acute injury literature. See Table 3. Systematic reviews that informed sport performance included studies that utilized both laser and LED components. LED parameters listed above are extracted from LED therapy-only studies from references [24,26].

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
