# Peer review of "Photobiomodulation as Medicine: Low-Level Laser Therapy (LLLT) for Acute Tissue Injury or Sport Performance Recovery"

_jfmk, 2024, doi:10.3390/jfmk9040181_

Round 1

Reviewer 1 Report

Comments and Suggestions for Authors

The authors present an interesting and useful review study about the use of photobiomodulation in the sport field, to heal wounds and to support athletic performances. I only have one main concern about Table 1: some of the assumptions or description are not perfectly pointing out the differences between lasers and LEDs. As an example the penetration depth of the light is depending not only on the light source, but also on its wavelength. The wavelength range for therapeutic lasers is larger: are the authors considering only the lasers used in LLLT? If this is the case, maybe they have to modify the Table 1 legend, that is mentioning therapeutic lasers in general. The same comments is for the Energy Density: are the authors sure that the indicated values for lasers and LEDs are covering all of the therapeutic applications, or is it limited to the LLLT applications? Exactly the same question for the treatment times. As concern Skin Application, usually the laser light is delivered through fibers, and in case of a LED can be delivered through fibers bundles; and normally a direct contact of the light source with the wounds is avoided. So the light dose is typically delivered through a non contact protocol. 

Another point is about Figure 1: it seems that it's showing the penetration depth of the 660 nm and of wavelengths longer than 760 nm. However, the legend is saying "Cellular mechanisms recruited by laser therapy may result in reduced inflammation and accelerated tissue recovery and/or healing", even if no cells, nor cell-cell interaction, or light-cell interation is depicted. 

At line 145 it is written Clincally, instead of Clinically.

At line 184 it is written msec, while the correct writing is ms. 

Author Response

Reviewer 1 comment 1: The authors present an interesting and useful review study about the use of photobiomodulation in the sport field, to heal wounds and to support athletic performances. I only have one main concern about Table 1: some of the assumptions or descriptions are not perfectly pointing out the differences between lasers and LEDs. As an example, the penetration depth of the light is depending not only on the light source, but also on its wavelength. The wavelength range for therapeutic lasers is larger: are the authors considering only the lasers used in LLLT? If this is the case, maybe they have to modify the table 1 legend, that is mentioning therapeutic lasers in general. The same comments is for the Energy Density: are the authors sure that the indicated values for lasers and LEDs are covering all of the therapeutic applications, or is it limited to the LLLY applications? Exactly the same question for the treatment times. As concern Skin Application, usually the laser light is delivered through fibres, and in case of a LED can be delivered through fibre bundles; and normally a direct contact of the light source with the wounds is avoided. So the light dose is typically delivered through a non-contact protocol.

Authors'  Response 1.1:   We appreciate the critical feedback regarding Table 1 (now Table 4 in the revised manuscript). For this Table, our aim is a high-level comparison of laser and LED parameters for therapeutic use in musculoskeletal context. Because of this, we have updated both the Table content and column titles for better clarity of the relevant variables. Next, we acknowledge that therapeutic "exposure" is dependent on several factors including wavelength and energy density/dose, because of this we went back to the published data to identify the specific ranges being used in our therapeutic areas of interest (sport performance/recovery, and acute injury rehabilitation). We edited the Table footnotes to clarify reference back to the summary tables (Tables 2 and 3). Lastly, we recognize that laser therapy would not be directly administered on a fresh wound and updated the text accordingly in the Table to note treatment at a distance.   We further reflected on the sequence of the manuscript text and moved Table 1 into the discussion section as the Table reflects what was found during the literature analysis and flows more appropriately with the discussion. With the adjusted sequence, the Table is now listed as Table 4 and is found in the Discussion section starting line 444.

Reviewer 1 comment 2: Another point is about Figure 1: it seems that it’s showing the penetration depth of the 660nm and of wavelengths longer than 760nm. However, the legend is saying “Cellular mechanisms recruited by laser therapy may result in reduced inflammation and accelerated tissue recovery and/or healing”, even if no cells, nor cell-cell interaction, or light-cell interaction is depicted.

Authors'  Response 1.2:  We agree with the reviewers input and elected to revise Figure 1 as an illustrative schematic (noted as not to scale in the figure legend).  Laser devices typically deliver specific wavelengths and are monochromatic, allowing the laser beam to penetrate deeper into the tissue compared to LEDs (see Table 4).  Power and energy density also differ between laser and LED devices, where lasers deliver light in a coherent and focused beam due to higher power density which supports laser light’s ability to penetrate deeply into tissues.  LED devices may offer therapeutic effects for cells and tissues closer to the skin surface.  Deeper penetration allows laser devices to target tissues and cells at greater depths in the body with the potential to recruit biological processes at a deeper level. Figure 1 is revised to illustrate this point.  

Reviewer 1 comment 3: At line 145 it is written Clincally, instead of Clinically.

Authors' Response 1.3: Agreed, we have updated the text accordingly.

Reviewer 1 comment 4: At line 184 is written msc, while the correct writing is ms.

Authors' Response 1.4: Agreed, we have updated the text accordingly

Reviewer 2 Report

Comments and Suggestions for Authors

Title: Photobiomodulation as Medicine: Low-level Laser Therapy (LLLT) for Acute Tissue Injury or Sport Performance Recovery

In this study, the authors investigated that LLLT helps in the treatment of acute tissue injuries and the recovery of sport performance. This is achieved by emitting low-level laser light, which penetrates tissues, stimulates biological processes such as promoting intracellular ATP production, reducing oxidative stress, and regulating inflammation. These biological effects contribute to faster healing, pain reduction, and enhanced post-exercise recovery, making it a clinically useful treatment method. This study is particularly valuable for sport physiotherapists, team physicians, trainers, and athletes who are interested in acute tissue injury treatment or sport performance recovery.

However, as this study is a review paper, the authors suggest conducting a meta-analysis with quantitative statistical analysis based on LLLT-related studies to derive more comprehensive conclusions.

Author Response

Reviewer 2 comment 1: The authors investigated that LLLT helps in the treatment of acute tissue injuries and the recovery of sport performance. This is achieved by emitting low-level laser light, which penetrates tissues, stimulates biological processes such as promoting intracellular ATP production, reducing oxidative stress, and regulating inflammation. These biological effects contribute to faster healing, pain reduction, and enhanced post-exercise recovery, making it a clinically useful treatment method. This study is particularly valuable for sport physiotherapists, team physicians, trainers, and athletes who are interested in acute tissue injury treatment or sport performance recovery. As this study is a review paper, the authors suggest conducting a meta-analysis with quantitative statistical analysis based on LLLT-related studies to derive more comprehensive conclusions.

Authors' Response 2.1:  We appreciate the Reviewer's feedback and elected to revise our text in the "Literature Search Strategy" section of the paper to strengthen our specific aims (starting approximately line 82) and strengthen our conclusions (line 540).  Our project considers the breadth of scientific evidence from the clinical literature rather than depth and aims to identify gaps in the overall state of research.  The purpose is to synthesize concepts and inform interdisciplinary teams of sport and health professionals working in athletic performance and/or recovery science.  As noted starting on Line 540 of the paper, a call to action from this publication would be the publication of guidelines for the profession to inform laser care alongside traditional therapeutic or conditioning protocols in sport and could be modeled after guidelines in other fields such as the International Consensus Guideline on Postsurgical and Traumatic Scarring [14] or WALT [9].  Such steps will serve to advance effective, safe and ethical use of PBM devices and advocate for improved regulations and standardization in this field for sport physiotherapists, team physicians, trainers, and athletes.

Round 2

Reviewer 1 Report

Comments and Suggestions for Authors

The manuscript has been improved and can be accepted for publication.